# Selection for feed efficiency elicits different postprandial plasma metabolite profiles in response to poor hygiene of housing conditions in growing pigs

Alícia Zem Fraga[1,2], Isabelle Louveau[2], Paulo Henrique Reis Furtado Campos[3], Luciano Hauschild[1], Nathalie Le Floc'h[2]*

1 Department of Animal Science, School of Agricultural and Veterinarian Sciences, São Paulo State University, Jaboticabal, São Paulo, Brazil, 2 PEGASE, INRAE, Institut Agro, Saint Gilles, France, 3 Department of Animal Science, Universidade Federal de Viçosa, Viçosa, Minas Gerais, Brazil

* nathalie.leffloch@inrae.fr

**Data Availability Statement:** The data underlying this study are available on the DataINRAE repository (https://doi.org/10.15454/U0RH9F).

## Abstract

This study was conducted to compare postprandial plasma concentrations of insulin, energy-related metabolites, and amino acids measured after a 6-week challenge consisting of exposure to good or poor hygiene of housing conditions of 24 growing pigs divergently selected for low-RFI (LRFI) and high-RFI (HRFI). Blood indicators of immune responses were assessed from samples collected before 0 (W0), and 3 (W3), and 6 weeks (W6) after pigs transfer to their respective hygiene of housing conditions. Plasma haptoglobin concentrations and blood neutrophil granulocyte numbers were greater in poor than in good hygiene of housing conditions at W3. Plasma concentrations of total immunoglobulin G were greater ($p = 0.04$) in poor than in good hygiene of housing conditions at W6. At W6, pigs were fitted with an intravenous catheter for serial blood samplings. Low-RFI pigs had greater insulin ($p < 0.001$) and lower triglyceride ($p = 0.04$) average plasma concentrations than HRFI pigs in both conditions. In poor hygiene of housing conditions, the peaks of insulin and glucose were observed earlier and that of insulin was greater in LRFI than in HRFI pigs. Irrespective of genetic line, average plasma concentrations of histidine, isoleucine, leucine, methionine, threonine, valine, and alanine were greater in poor compared with good hygiene of housing conditions. Only HRFI pigs had greater lysine, asparagine, proline, and tyrosine plasma concentrations in poor than in good hygiene of housing conditions. Conversely, arginine, tryptophan, proline, and tyrosine plasma concentrations were lower only for LRFI pigs housed in poor hygiene conditions. Our results suggest that, contrary to HRFI, LRFI pigs increase or maintain their utilization of tryptophan, arginine, and lysine when housed in poor hygiene conditions. This indicates that this difference may contribute to the better capacity of LRFI to cope with poor hygiene of housing conditions.

**Funding:** This study was supported by the European Union's Seventh Framework Programme for Research, Technological Development and Demonstration (grant number 613574, PROHEALTH project). Alícia Zem Fraga was supported by a scholarship from the São Paulo Research Foundation (FAPESP-Brazil; grant number 2018/15559-7 and fellowship number 2018/11807-6).

**Competing interests:** The authors have declared that no competing interests exist.

## Introduction

Selection of pigs for residual feed intake (RFI) has been used to improve feed efficiency. Briefly, RFI is the difference between the observed feed consumption of an animal and that predicted by the estimated requirements for maintenance and growth of a reference population. High-RFI (HRFI) pigs eat more than predicted and therefore are less efficient than low-RFI (LRFI) pigs [1]. Difference in feed efficiency between RFI lines is explained by changes in physical activity [2], heat production [3], and in the partition of nutrients between maintenance and growth [4]. Difference in nutrient partitioning is suspected to alter pig ability to allocate nutrients for stress and immune responses when facing environmental challenges [5]. In commercial farms, pigs are often exposed to stressful situations such as weaning, mixing, high stocking density, transport, and poor hygiene of housing conditions resulting in immune system activation. In turn, immune activation, including inflammation, results in changes in nutrient metabolism, which therefore reduces nutrient availability for growth [6].

The better ability of LRFI compared with HRFI growing pigs to cope with an immune challenge caused by poor hygiene of housing conditions was previously reported [7]. More specifically, we showed that after being housed six weeks in poor hygiene conditions, growth performance and health, assed by lung lesion scoring and blood indicators, were more affected in HRFI pigs than in LRFI pigs. We hypothesized that this difference in coping ability between RFI pigs may involve changes in their metabolism after a 6-week exposure to contrasted hygiene of housing conditions. To evaluate this metabolic response, we analysed, on a subset of pigs from the study by Chatelet et al. 2018 [7], the patterns of plasma postprandial concentrations of insulin, energy-related metabolites, urea and amino acids (AA), as it has been shown previously that they reflect changes in the use of nutrients for anabolic and catabolic processes. Indeed, the same methodology was recently reported to describe the metabolic status of growing pigs in response to an inflammatory challenge and high ambient temperature [8] or to compare castrated and entire male pigs [9]. Therefore, the present study was carried out to compare, in LRFI and HRFI pigs, the effects of an exposure to poor hygiene of housing conditions for six weeks on pre- and postprandial plasma concentrations of insulin, energy-related metabolites, urea and free amino acids.

## Material and methods

The experiment was conducted at INRAE UE3P (Saint-Gilles, France) in accordance with the ethical standards of the European Community (Directive 2010/63/EU), and was approved by the local ethical committee (Comité rennais d'éthique en matière d'expérimentation animale or CREEA N˚ 07). The experiment approval number is APAFIS#494–2015082717314985.

### Animals, diets, and experimental design

The experiment was conducted on a subset of 24 Large-White pigs from a larger study (n = 160 pigs) previously described [7]. Animals with representative body weight (BW) of each experimental group (see below) were selected at 12 weeks of age. Selected pigs were fed, housed, and submitted to the same experimental procedures as the whole set of pigs, before being involved in the serial blood sampling 6 weeks later.

Pigs originated from the 8th generation of a selection program for divergent RFI conducted at INRAE. Briefly, the lines were established using the RFI selection criterion between 35 and 95 kg BW, calculated as: RFI = ADFI − (1.24 × ADG) − (31.9 × BFT), where ADFI was the average daily feed intake (g/day), ADG the average daily gain (g/day) and BFT was the backfat thickness (mm) at 95 kg [10]. The study was performed as a 2 × 2 factorial design including four experimental groups: HRFI and LRFI pigs housed in good hygiene conditions (good-

HRFI, good-LRFI); and HRFI and LRFI pigs housed in poor hygiene conditions (poor-HRFI, poor-LRFI). Briefly, poor hygiene of housing conditions consisted of no cleaning nor sanitation of the room after the previous occupation by non-experimental pigs [7]. In contrast, good hygiene of housing conditions included room cleaning, disinfection, and adoption of strict biosecurity precautions throughout the experimental period. The staff wore clean boots, clothes and gloves before entering the room.

According to their allocation, pigs were placed in one of the two experimental rooms (good or poor hygiene of housing conditions) for 6 weeks. In each room, pigs were housed in individual concrete floor pens (85 × 265 cm) equipped with a feed dispenser and a nipple drinker. Pigs had free access to water and were fed *ad libitum* a standard diet formulated to meet the nutritional requirement of growing pigs (Table 1). After 6 weeks, the 24 selected pigs (n = 6 per experimental group) were fitted with an intravenous ear catheter following a minimally invasive procedure. Briefly, after an overnight fast, pigs were premedicated with 15 mg/kg of ketamine injected intramuscularly (Imalgène 1000, Merial, Lyon, France) and were then anesthetized by inhalation of sevoflurane (Sevoflurane, Baxter, Maurepas, France) using a facemask. An intravenous catheter was inserted through a small incision on the flap of the ear. The external part of the catheter was fixed on the ear skin and a connector was added for blood samplings the day after catheter insertion. No drug was used to avoid interference with the health status of pigs.

**Table 1. Ingredients and chemical composition of the diet.**

| Items | Standard diet |
|---|---|
| Ingredient composition as-fed basis, % | |
| Wheat | 32.17 |
| Corn | 15.00 |
| Barley | 24.95 |
| Wheat bran | 5.00 |
| Rapeseed meal | 7.00 |
| Soybean meal | 11.47 |
| Vegetable oil | 1.00 |
| Calcium carbonate | 1.51 |
| Dicalcium phosphate | 0.10 |
| Salt | 0.45 |
| Liquid lysine (50.0% of L-Lysine) | 0.53 |
| DL-Methionine | 0.04 |
| L-Threonine | 0.10 |
| L-Tryptophan | 0.07 |
| Vitamin and mineral premix[1] | 0.50 |
| Phytase and organic acids | 0.11 |
| Chemical composition | |
| ME, MJ/kg | 12.64 |
| NE, MJ/kg | 9.48 |
| CP, % | 15.50 |

[1] Mineral vitamin supplement (per kg of diet): Vit. A (1.000.000 UI); Vit. D3 (200.000 UI); Vit. E (4.000 UI); Vit. K3 (400 mg); Vit. B1 (400 mg); Vit. B2 (800 mg); Vit. B6 (200 mg); Niacin (3.000 mg); Pantothenic acid (2.000 mg); Folic acid (200 mg); Biotin (40 mg); Vit. B12 (4.4 mg); Copper (2.000 mg); Iodine (40 mg); Manganese (8.000 mg); Selenium (30 mg); Zinc (20.000 mg); Ferrous sulphate (11.200 mg); Ferrous carbonate (4.800 mg); and Choline chloride (100.000 mg).

## Blood sample collection

Blood samples were collected at fasted state by jugular vein puncture at week zero (W0; before pig transfer to the respective hygiene conditions) and week three (W3), and from the intravenous ear catheter at week six (W6). These samples were used to measure blood indicators of immune responses (immunoglobulin G (IgG), haptoglobin, and number of blood neutrophil granulocytes). Serial blood samplings were performed at W6 the day after catheter insertion. After being fasted overnight, each pig was offered 300 g of the standard diet at 08-h, irrespective of the hygiene conditions and the genetic line. This meal size was determined to ensure that all pigs eat their meal in less than 10 min [9] and corresponded to 15% of the ADFI. Blood samples (6 ml) were collected from the catheter before the meal delivery (fasted state; t0), and then at 15, 30, 45, 60, 75, 90, 105, 120, 150, 180, 210, and 240 min after the meal delivery. Blood samples were collected on ethylenediaminetetraacetic acid (EDTA) tubes for insulin, glucose, free fatty acid (FFA), triglyceride, and urea measurements; and on heparinized tubes for AA analyses. Samples were immediately placed on ice, except for EDTA tubes used for measuring the number of blood neutrophil granulocytes, and then centrifuged ($1800 \times g$) for 10 min at 4°C. Plasma was collected and stored at -80°C for AA, and at -20°C for other plasma parameters.

## Blood cell and plasma variable analyses

The number of blood neutrophil granulocytes was measured with a haematology automated cell counter calibrated for pigs (MS9; Melet Schloesing Laboratories, Osny, France). Quantitative sandwich ELISA tests were used to quantify total IgG plasma concentrations [11]. Plasma concentrations of haptoglobin (phase haptoglobin assay T801; Tridelta Development Ltd, Maynooth, Ireland), glucose (Kit Glucose RTU, ref. 61269; Biomérieux, Marcy-l'Etoile, France), and free fatty acids (Kit WAKO NEFA; Sobioda, Montbonnot-Saint-Martin, France), were analysed by an automated enzymatic method using commercial kits and a multianalyzer apparatus (Konelab 20i, ThermoFisher Scientific, Courtaboeuf, France). For plasma triglyceride and urea concentration measurements, kits were obtained from Thermo Fisher Diagnostics SAS (Asnieres-Sur-Seine, France). Plasma insulin concentrations were determined using a commercial immunoassay kit (ST AIA-PACK IRI) and the AIA-1800 device (Automated Immunoassay Analyzer; TOSOH Bioscience, Tokyo, Japan). Assay sensitivity for insulin was 0.5 µUI/ml and the intra-assay CV was below 5%. Plasma free AA concentrations were determined by an ultra-performance liquid chromatography (UPLC) apparatus (Waters Acquity Ultra Performance LC, Waters, Milford, MA, USA) after derivatization of samples using the AccQ Tag Ultra method (MassTrak AAA; Waters, Milford, MA, USA) as previously described [12].

## Calculations and statistical analysis

The pig was the experimental unit. Average plasma concentrations of insulin, energy metabolites, and AA were calculated from pre and postprandial concentrations. Average concentrations and concentrations at each sampling time were analysed using the linear MIXED procedure (SAS Inst. Inc., Cary, NC). The model included the genetic lines (LRFI or HRFI), hygiene of housing conditions (poor or good), sampling time (time), and their interactions as fixed effects. The repeated measurements option was used with a compound symmetry covariance structure to account for animal effect over sampling time. Adjusted means were compared using the Bonferroni test. Plasma concentrations of total IgG and haptoglobin, and number of neutrophil granulocytes were compared between the two hygiene conditions with a

non-parametric test (Median test) using the NPAR1WAY procedure of SAS. Probabilities less than 0.05 were considered significant.

For indispensable AA, when an interaction between genetic line and hygiene of housing conditions was significant, plasma profiles were analysed by nonlinear regression using a one-compartment model with Erlang retention times [13] in combination with a constant basal concentration. The Erlang distribution was previously described by [14]. The model used to describe the change in AA concentrations over time as an asymmetric bell-shaped curve was:

$$\text{Amino acid concentration } C(t) = \frac{k(\lambda^n \times (t)^{n-1}) \times \exp(-\lambda \times t) + C\text{basal}}{(n-1)!}$$

where $C$basal is the basal concentration, $k$ is a scale parameter, $\lambda$ and $n$ are shape parameters of the Erlang distribution of residence times, and $t$ is the time. For each AA plasma concentration curve, the shape parameter $n$ was tested using values ranging from one to four, and the result with the lowest residual SD was retained. The model was parameterized to obtain AA concentration at $t = 0$ ($C_0$), the maximum AA concentration ($C$max), and the time when AA concentration is maximum ($T$max). To test whether the parameters of the model differed between the experimental groups, a sum of squares reduction test was used [15]. As our main objective was to compare the effects of the hygiene challenge in LRFI and HRFI pigs, this test was run to compare poor-LRFI to good-LRFI, and poor-HRFI to good-HRFI. The principle was to compare a "full model" where all parameters differ between the two experimental groups with a "reduced model", which has common parameters. This was done for each parameter successively. An F-test was used to test if the models were statistically different and a probability less than 0.05 was considered as significant.

## Results

### General observations

One good-LRFI pig was excluded from analyses because it did not fully consume the 300 g of feed during the allocated time. Growth performance and blood indicators of immune responses are presented in Fig 1. Pig initial and final body weights were 26.5 ± 2.80 kg and 55.4 ± 6.88 kg, respectively (Fig 1A). A numeric but no significant effect of hygiene of housing conditions was reported for ADG from W0 to W6 (0.683 vs. 0.614 kg/day for good and poor hygiene of housing conditions, respectively; $p = 0.12$). No difference between pigs allotted to poor or good hygiene of housing conditions were reported for any measured variables at W0 ($p > 0.05$). At W3, plasma haptoglobin concentrations and blood neutrophil granulocyte counts were greater in pigs housed in poor than in good hygiene conditions ($p < 0.05$; Fig 1B and 1C). At W6, total IgG plasma concentrations were greater in pigs housed in poor hygiene conditions compared with pigs housed in good hygiene conditions ($p = 0.04$; Fig 1D).

### Plasma concentrations of insulin, energy-related metabolites and urea

Average concentrations of insulin, energy-related metabolites, and urea are presented in Table 2. Plasma fasted concentrations did not differ for any measured variables ($p > 0.05$). There was a sampling time effect ($p < 0.05$) for all metabolites studied. No hygiene effect and interaction between Line × Hygiene were reported ($p > 0.05$). Regardless of hygiene conditions, LRFI pigs had greater insulin ($p < 0.001$) and lower average plasma concentrations of triglycerides ($p = 0.04$) than HRFI pigs.

The Line × Hygiene × Time interaction was significant for insulin, glucose, and triglycerides (Fig 2). In poor hygiene of housing conditions only, the maximum insulin concentration

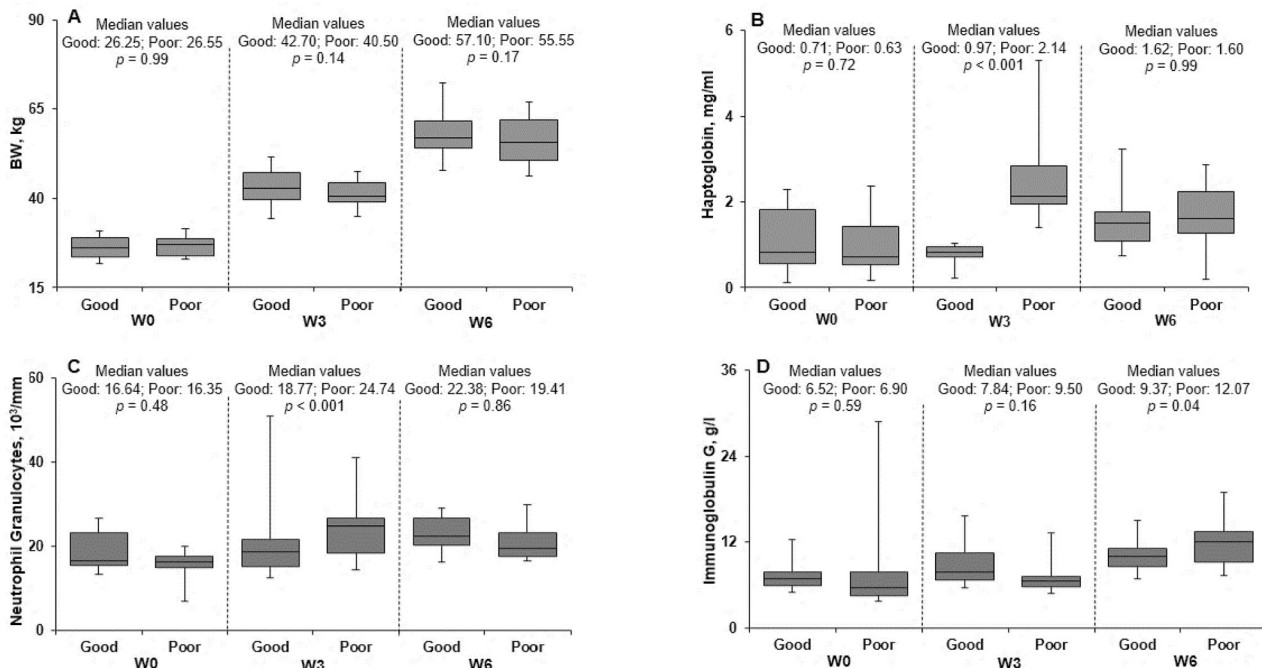

**Fig 1. Body weight (A) and blood indicators of immune and inflammatory responses [(B) haptoglobin, (C) neutrophil granulocytes, and (D) immunoglobulin G] before 0 (W0), and 3 (W3) and 6 weeks (W6) after pigs transfer to good or poor hygiene of housing conditions.**

occurred earlier (45 min) and was greater in LRFI compared with HRFI pigs ($p < 0.001$; Fig 2A). For glucose, the peak value did not differ between lines ($p = 0.98$) but occurred earlier (30 min) in LRFI than in HRFI pigs when housed in poor hygiene conditions ($p < 0.001$; Fig 2B). Regarding plasma triglyceride concentrations, the minimum concentration did not differ between the four experimental groups but occurred earlier (45 min) in HRFI pigs housed in poor hygiene conditions compared with the three other groups (75 min; Fig 2D).

**Table 2. Average plasma concentrations of insulin, energy-related metabolites, and urea measured in low and high residual feed intake pigs (LRFI and HRFI) housed in good (good) or poor (poor) hygiene conditions at week 6.**

| | LRFI | | HRFI | | | p-value[4] | | |
|---|---|---|---|---|---|---|---|---|
| | Good | Poor | Good | Poor | SEM | Line | Hyg | Line×Hyg |
| No.[1] | 5 | 6 | 6 | 6 | | | | |
| Average plasma concentrations[2] | | | | | | | | |
| Insulin, mU/l | 27.8 | 29.3 | 17.3 | 15.3 | 1.87 | <0.001 | 0.89 | 0.37 |
| Glucose, mg/l | 1050 | 1044 | 1011 | 1009 | 22 | 0.11 | 0.86 | 0.93 |
| FFA[3], µmol/l | 99.7 | 125.3 | 134.8 | 107.3 | 23.8 | 0.72 | 0.97 | 0.28 |
| Triglycerides, mg/l | 274 | 262 | 343 | 321 | 30 | 0.04 | 0.58 | 0.87 |
| Urea, mg/l | 140 | 150 | 143 | 158 | 13 | 0.70 | 0.36 | 0.86 |

[1]No. = number of animals per group.

[2]Average plasma concentrations include fasted and postprandial concentrations measured for 4-h after ingestion of 300 g of feed. There was no effect of the experimental treatments on fasted concentrations for any blood variables ($p > 0.05$).

[3]FFA = free fatty acids.

[4]Probability values for the effect of genetic lines (Line), hygiene conditions (Hyg), and their interaction. There was an effect of sampling time ($p < 0.05$) for all variables studied. The interactions with time are presented in Fig 2.

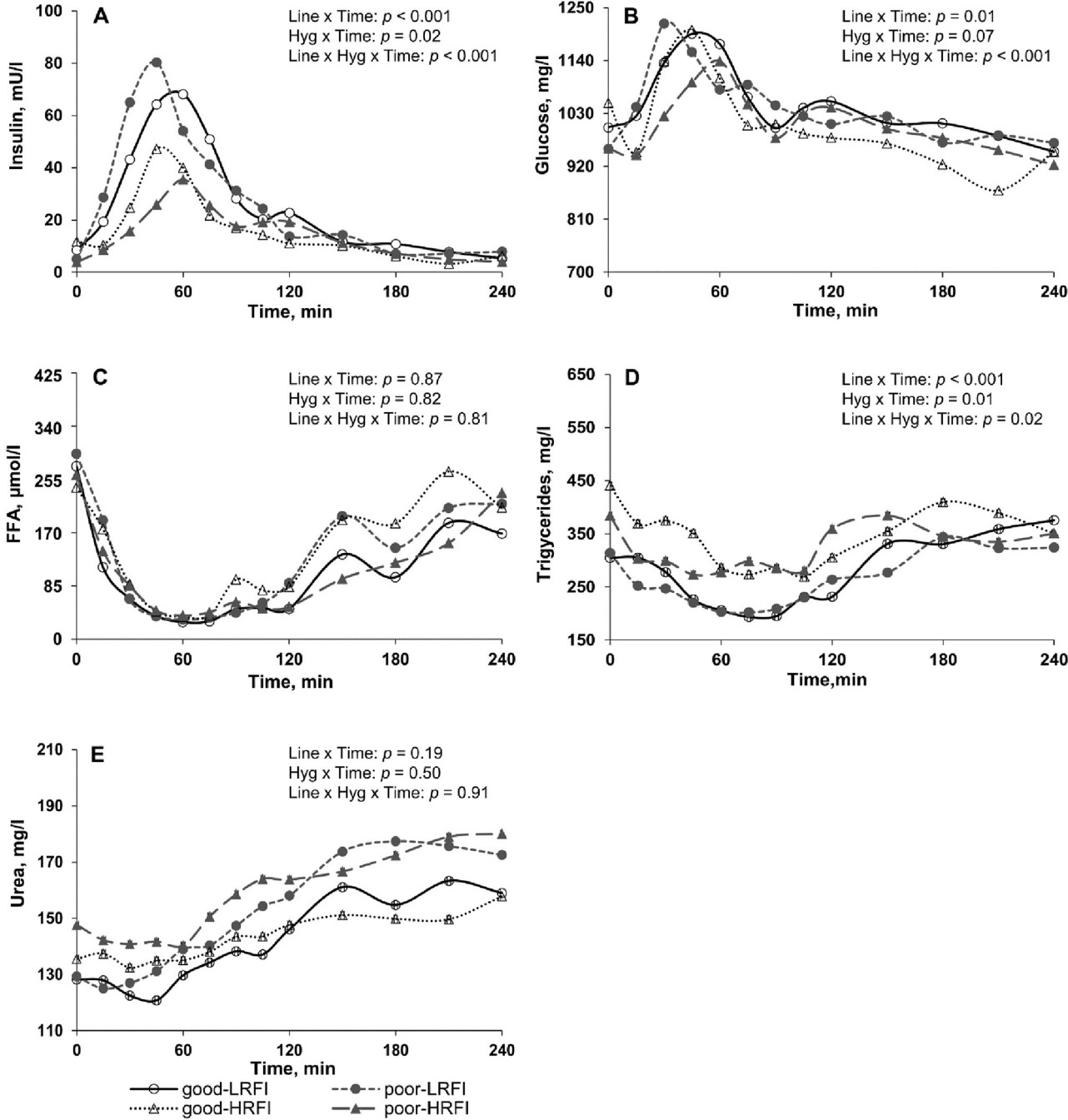

**Fig 2. Postprandial plasma profiles of (A) insulin and metabolites [(B) glucose, (C) free fatty acids; FFA, (D) triglycerides, and (E) urea] measured in low and high residual feed intake pigs (LRFI and HRFI) housed in good or poor hygiene conditions at week 6.**

### Free plasma AA average concentrations

Average concentrations of plasma free AA are presented in Table 3. Except for cysteine (Cys), there was a sampling time effect ($p < 0.05$) for all AA. Regardless of hygiene of housing conditions, LRFI pigs had greater Met, Thr, asparagine (Asn), Cys, glutamine (Gln), glutamate

**Table 3. Average concentrations of plasma free AA (nmol/ml) measured in low and high residual feed intake pigs (LRFI and HRFI) housed in good (good) or poor (poor) hygiene conditions at week 6.**

| | LRFI | | HRFI | | | $p$-value[3] | | |
|---|---|---|---|---|---|---|---|---|
| | **Good** | **Poor** | **Good** | **Poor** | **SEM** | **Line** | **Hyg** | **Line×Hyg** |
| No.[1] | 5 | 6 | 6 | 6 | | | | |
| Average plasma AA concentrations[2] | | | | | | | | |
| Arginine | 151.5[a] | 139.2[b] | 100.3[c] | 99.8[c] | 8 | <0.001 | 0.09 | 0.04 |
| Histidine | 70.3 | 69.3 | 75.8 | 82.8 | 10 | <0.001 | 0.02 | 0.05 |
| Isoleucine | 122.2 | 135.2 | 127.6 | 140.0 | 19 | 0.12 | <0.001 | 0.08 |
| Leucine | 174.7 | 186.6 | 177.6 | 191.3 | 12 | 0.13 | <0.001 | 0.10 |
| Lysine | 243.8[a] | 234.5[ab] | 216.9[b] | 246.2[a] | 15 | 0.16 | 0.02 | <0.001 |
| Methionine | 40.0 | 45.9 | 30.6 | 36.1 | 11 | <0.001 | <0.001 | 0.95 |
| Phenylalanine | 83.2 | 82.3 | 84.5 | 85.8 | 8 | 0.84 | 0.14 | 0.58 |
| Threonine | 120.1 | 129.9 | 84.0 | 95.0 | 17 | <0.001 | <0.001 | 0.44 |
| Tryptophan | 62.5[a] | 51.7[b] | 44.2[bc] | 41.4[c] | 37 | <0.001 | <0.001 | <0.001 |
| Valine | 250.4 | 267.8 | 260.7 | 279.9 | 15 | 0.04 | <0.001 | 0.32 |
| Alanine | 448.3 | 496.4 | 527.0 | 564.5 | 3 | <0.001 | <0.001 | 0.39 |
| Asparagine | 60.3 | 59.1 | 52.3 | 54.1 | 17 | <0.001 | 0.92 | 0.99 |
| Aspartate | 18.2[a] | 18.4[a] | 13.8[c] | 15.2[b] | 5 | <0.001 | 0.16 | <0.001 |
| Cystine | 40.4 | 41.0 | 36.9 | 38.2 | 10 | 0.04 | 0.20 | 0.66 |
| Glutamine | 592.2 | 590.1 | 522.4 | 517.8 | 7 | <0.001 | 0.88 | 0.64 |
| Glutamate | 207.5 | 214.5 | 157.8 | 162.4 | 6 | <0.001 | 0.53 | 0.11 |
| Glycine | 767.2 | 733.9 | 732.0 | 713.0 | 12 | 0.01 | 0.02 | 0.55 |
| Proline | 302.1[a] | 288.6[b] | 248.5[d] | 274.7[c] | 14 | <0.001 | 0.06 | 0.02 |
| Serine | 134.1[b] | 141.6[a] | 125.3[b] | 133.1[b] | 9 | <0.001 | <0.001 | <0.001 |
| Tyrosine | 71.0[a] | 56.9[b] | 48.6[c] | 58.9[b] | 16 | <0.001 | 0.46 | <0.001 |

[1]No. = number of animals per group.

[2]Average plasma concentrations include fasted and postprandial concentrations measured for 4-h after ingestion of 300 g of feed.

[3]Probability values for the effect of genetic lines (Line), hygiene conditions (Hyg), and their interaction. Except for cystine ($p = 0.18$), there was an effect of sampling time ($p < 0.05$) for all AA studied. The parameters of the model for indispensable AA whose Line×Hyg interaction was significant are presented in Table 4. There was Line x Time effect for Arg, only ($p < 0.001$).

[a,b,c] Within a row values with different superscripts differed ($p < 0.05$).

(Glu), and glycine (Gly) ($p < 0.05$), and lower histidine (His), valine (Val), and alanine (Ala) plasma concentrations ($p < 0.05$) than HRFI pigs. Irrespective of genetic line, average plasma concentrations of His, isoleucine (Ile), leucine (Leu), Met, Thr, Val, and Ala were greater ($p < 0.05$) and Gly were lower ($p = 0.02$) in poor than in good hygiene of housing conditions. For dispensable AA, the interaction between Line and Hygiene was significant for aspartate (Asp), proline (Pro), serine (Ser), and tyrosine (Tyr). Average concentrations of Asp were greater in poor than in good hygiene of housing conditions in HRFI pigs only ($p < 0.001$). When housed in poor hygiene conditions, LRFI pigs had lower and HRFI had greater plasma concentrations of Pro ($p < 0.05$) than in good hygiene conditions. Low-RFI pigs had greater average concentrations of Ser in poor than in good hygiene of housing conditions ($p < 0.001$) whereas average concentrations did not differ in HRFI pigs ($p = 0.72$). Poor hygiene of housing conditions resulted in lower average Tyr concentrations in LRFI and greater in HRFI pigs compared with good conditions ($p < 0.001$). For indispensable AA, the interaction between Line and Hygiene was significant for arginine (Arg), Lys and Trp. These results are described in details in the following paragraph.

**Table 4. Values of parameters describing the postprandial kinetics of plasma free arginine, lysine, and threonine in low and high residual feed intake pigs (LRFI and HRFI) housed in good (good) or poor (poor) hygiene conditions at week 6.**

| | LRFI | | P-value[3] | HRFI | | P-value[3] |
|---|---|---|---|---|---|---|
| | **Good** | **Poor** | | **Good** | **Poor** | |
| No.[1] | 5 | 6 | | 6 | 6 | |
| Parameter values[2] | | | | | | |
| Arginine | | | | | | |
| C0, μM | 88 ± 6 | 69 ± 5 | 0.02 | 62 ± 6 | 53 ± 6 | 0.28 |
| Cmax, μM | 193 ± 12 | 188 ± 7 | 0.62 | 125 ± 5 | 130 ± 3 | 0.75 |
| Tmax, min | 86 ± 4 | 81 ± 2 | 0.84 | 87 ± 5 | 96 ± 3 | 0.27 |
| Lysine | | | | | | |
| C0, μM | 126 ± 10 | 127 ± 8 | 0.98 | 145 ± 9 | 132 ± 9 | 0.34 |
| Cmax, μM | 349 ± 2 | 329 ± 7 | 0.12 | 273 ± 5 | 315 ± 3 | <0.001 |
| Tmax, min | 77 ± 3 | 69 ± 2 | 0.42 | 74 ± 7 | 86 ± 3 | 0.61 |
| Tryptophan | | | | | | |
| C0, μM | 39 ± 2 | 27 ± 2 | <0.001 | 25 ± 2 | 22 ± 2 | 0.79 |
| Cmax, μM | 72 ± 0 | 63 ± 4 | 0.17 | 52 ± 2 | 50 ± 7 | 0.88 |
| Tmax, min | 87 ± 3 | 68 ± 5 | <0.001 | 91 ± 6 | 87 ± 4 | 0.29 |

[1]No. = number of animals per group.

[2]Values are means ± SD. C0: the concentration at t = 0, Cmax: the maximum concentration, Tmax: the time in minute at maximum concentration.

[3]Probability values for the effect of hygiene conditions.

## Plasma free indispensable AA postprandial profiles

Average Arg concentrations were lower in poor than in good hygiene of housing conditions in LRFI pigs only ($p = 0.04$; Table 4). This effect was associated with lower C0 value in LRFI pigs when housed in poor hygiene conditions ($p = 0.02$; Table 4 and Fig 3A). Lysine average concentrations ($p = 0.02$; Table 4) and Cmax ($p < 0.001$; Table 4 and Fig 3B) value were higher in poor hygiene of housing conditions in HRFI pigs only. Lower Trp concentrations were observed in poor hygiene of housing conditions in LRFI pigs only ($p < 0.001$; Table 4). For LRFI pigs, the values of C0, and Tmax for Trp were lower in poor than in good hygiene of housing conditions ($p < 0.001$; Table 4 and Fig 3C).

## Discussion

This study was conducted to compare the metabolic modifications induced by poor hygiene of housing conditions in LRFI and HRFI growing pigs. Our major and original finding is that the effects of the poor hygiene of housing conditions on the postprandial profiles of insulin and some indispensable AA differ between the two RFI lines. We discuss if such differences may explain the better coping ability of LRFI pigs previously reported [7].

To compare metabolic changes induced by the experimental factors, i.e. to compare the response of HRFI and RFI pigs housed in poor or good hygiene of housing conditions for six weeks, we analysed the average postprandial plasma concentrations of insulin, energy-related metabolites, urea, and amino acids, as well as the pattern of postprandial plasma indispensable AA profiles. Briefly, after being fasted overnight, on the day of serial blood sampling, all pigs received 300 g of the same feed that they consumed in less than 10 min and blood was collected at fasted state and then, for 4 hours after the meal. Thus, differences in plasma profiles were associated to differences in both digestion and postprandial metabolism induced by the

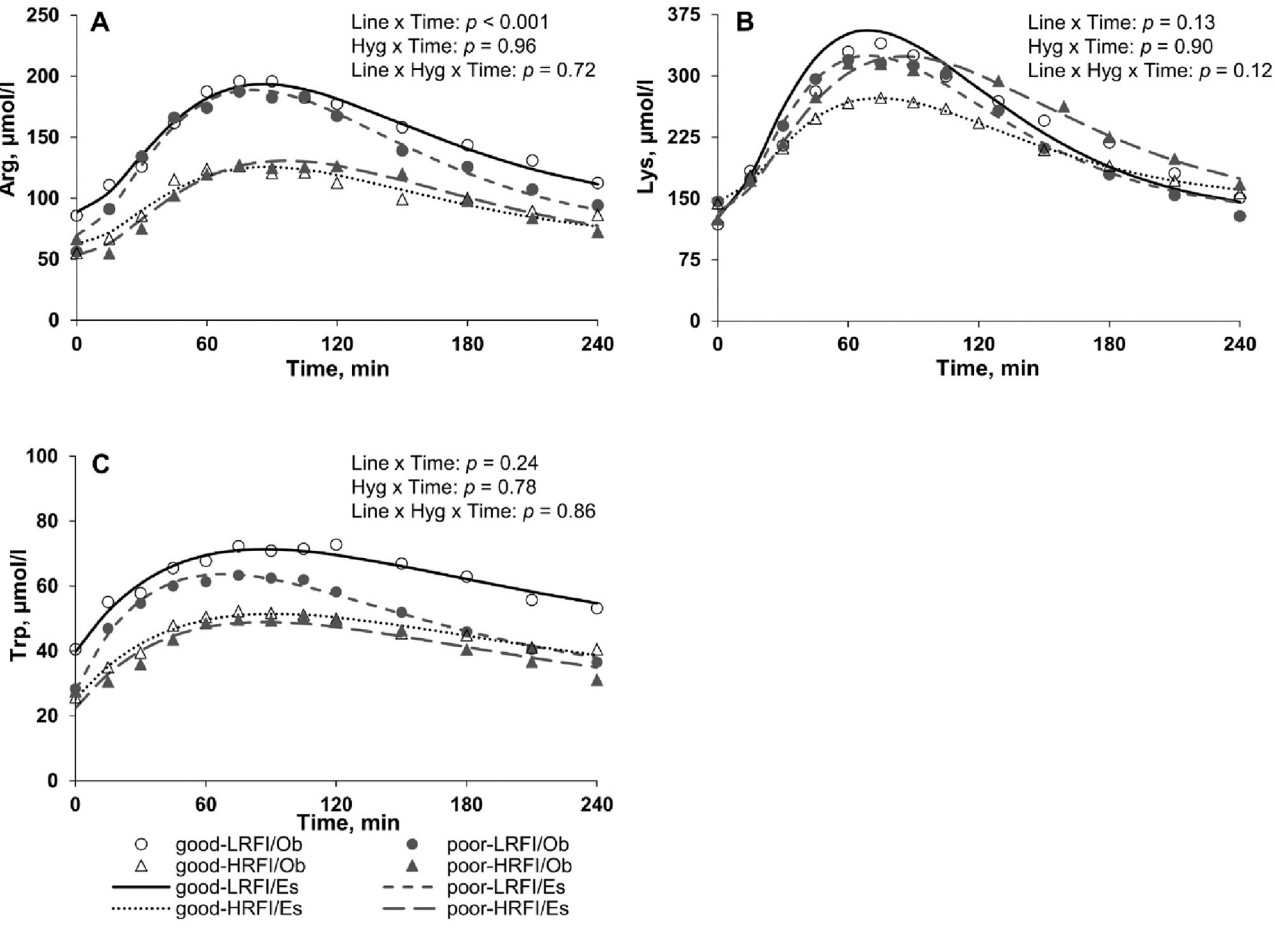

**Fig 3. Plasma profiles of free indispensable amino acids [(A) arginine; Arg, (B) lysine; Lys, and (C) tryptophan; Trp] measured in low and high residual feed intake pigs (LRFI and HRFI) housed in good or poor hygiene conditions at week 6.** The observed (Ob) and estimated (Es) values are presented.

experimental factors, namely the genetic line, the hygiene of housing condition, and their interaction. In the current study, a model of poor hygiene of housing conditions was applied for 6 weeks to induce an immune system activation. The findings of greater plasma haptoglobin concentrations and blood neutrophil granulocyte counts in poor than in good hygiene of housing conditions at W3 indicate that the model successfully induced an inflammation and activated immune responses [16]. Indeed, haptoglobin a major acute phase protein in pig, and blood neutrophil granulocyte counts are both relevant indicators of inflammation in pigs [17, 18]. Moreover, greater total IgG plasma concentrations in poor hygiene of housing conditions at W6 show that the immune system was still stimulated at the time of serial blood sampling. Contrary to AA, energy-related metabolites and insulin did not differ between poor and good hygiene of housing conditions. It should be noted that in our study, insulin and energy metabolite concentrations were measured 6 weeks after the beginning of the challenge when pigs were probably recovering from inflammation as indicated by lower blood haptoglobin concentrations at W6 compared with W3. Indeed, during an inflammatory challenge, plasma glucose concentrations were restored two days after being temporarily increased in young growing pigs [12] showing the fast return to glucose homeostasis. Pigs housed in poor hygiene

conditions had greater concentrations of Ala, His, Met, Thr, and branched-chain AA (BCAA) than pigs housed in good conditions. Greater AA plasma concentrations may be due to a lower AA retention as muscle protein in immune challenged pigs [19] caused by an increase in protein breakdown and/or decrease in protein synthesis. If our experiment conducted on a low number of pigs failed to report a significant reduction in ADG in response to poor hygiene, growth performance measured in the whole set of pigs was depressed after 6 weeks of housing in poor hygiene conditions and this effect was mainly caused by the inflammation and the activation of the immune system [7]. Conversely, lower Thr postprandial plasma concentrations were reported in pigs coinfected with *Mycoplasma hyopneumoniae* and influenza virus probably to support a great demand for immunoglobulin synthesis [11]. Such a discrepancy regarding the response of Thr concentrations in our study was unexpected since the hygiene challenge increased immunoglobulin plasma concentrations but this increase was probably too moderate to affect plasma Thr concentrations. From these results, it can be suggested that the immune system activation impacted glucose metabolism more rapidly and for a shorter period than protein metabolism, as previously reported after a change in hormonal status in growing pigs [9]. Accordingly, in septic rats, plasma concentrations of tumor necrosis factor α measured 1.5 hours after an experimental infection showed a strong correlation with changes in protein metabolism and body weight two weeks later [20], demonstrating that prolonged effects on nitrogen metabolism may be observed while blood indicators are no more detectable or return to normal values.

Contrary to the hygiene challenge that affects metabolism and digestibility [18], the selection for RFI did not affect the digestibility of a standard low fiber feed [21]. Thus, differences in plasma profile between RFI lines are mostly explained by a difference in metabolism. Irrespective of hygiene conditions, LRFI pigs had greater average plasma concentrations of insulin and lower plasma concentrations of triglycerides than HRFI pigs. In agreement with our findings, Montagne et al. [22] observed greater plasma insulin concentrations after the ingestion of a small meal and Le Naou et al. [23] reported lower plasma triglyceride concentrations measured at fed state in LRFI compared with HRFI pigs. In pigs, plasma triglycerides result from the lipolysis of lipids stored in adipose tissue. Insulin is an anabolic hormone with a potent anti-lipolytic action on adipose tissue [24]. Therefore, lower triglyceride concentrations may be partly explained by the greater insulin concentrations in plasma. Alternatively, lower plasma triglycerides levels may also be attributed to the lower body fat content of LRFI compared with HRFI pigs [25]. Moreover, the impact of the overnight fasting associated with feed restriction on the day of serial blood samplings may be greater in HRFI pigs forcing them to mobilize their body fat reserve. However, feed restriction of HRFI pigs did not affect their plasma triglyceride concentrations [23]. Poor hygiene of housing conditions impacted differently insulin response in LRFI and HRFI. Indeed, the postprandial peak of insulin occurred earlier and was greater in LRFI compared with HRFI pigs when housed in poor hygiene conditions. Such an effect contributes to explain why LRFI pigs maintained their growth rate in poor hygiene of housing conditions [7]. Insulin is indeed the main hormone allowing the postprandial AA utilization for protein synthesis in muscle [26].

Plasma urea concentration did not significantly differ between the two lines. Urea is produced from the deamination of AA and, at fed state, reflects the catabolism of dietary AA that are not used for body protein synthesis and deposition. Our result is in line with a previous study showing that genetic selection for RFI did not affect nitrogen metabolism when HRFI and LRFI pigs were fed the same restricted level of feed [4]. However, differences in plasma free AA average concentrations were observed between the two RFI lines. For instance, His, Val, and Ala plasma concentrations were lower in LRFI than in HRFI pigs. Lower plasma Ala concentrations at fed state were previously reported in LRFI compared with HRFI pigs [12]

suggesting a lower muscular release of Ala for hepatic glucose synthesis (Cahill cycle) in LRFI pigs. This is in accordance with lower energy expenditure in LRFI than in HRFI pigs [4]. The BCAA (Ile, Leu and Val) are the major donors of the amino group for the synthesis of Ala from pyruvate in muscle [12]. In the present experiment, despite that only Val concentrations differed between the two lines, Ile and Leu plasma concentrations were numerically lower in LRFI than in HRFI pigs. Alanine synthesis in muscle and its release in the plasma in LRFI pigs may have been reduced by the decreased availability of BCAA. In the current study, Arg and Trp average and basal plasma concentrations were lower in LRFI pigs when housed in poor compared with good hygiene conditions whereas the challenge did not affect plasma concentrations of these two AA in HRFI pigs. Besides being proteinogenic AA, both Trp and Arg are known to be involved in immune-related metabolic pathways. During immune activation, Trp is catabolized in kynurenine by the indoleamine 2,3-dioxygenase (IDO) enzyme, a metabolic pathway involved in the regulation of immune responses [27]. Arginine is an AA serving as a precursor for the synthesis of polyamines that is massively used by rapidly dividing cells like proliferating lymphocytes [28]. It is also involved in the synthesis of creatine that has anti-oxidative and anti-inflammatory functions [29]. Lower C0 values for Trp and Arg may reflect an effect of immune activation that is independent of postprandial use of AA for muscle anabolism. Besides, for Trp, the time-related variations differed between hygiene of housing conditions in LRFI pigs with earlier time at maximum plasma concentration (Tmax) in poor than those housed in good hygiene conditions. Such results might be a consequence of faster postprandial clearance of dietary Trp for both immune responses and muscle anabolism. If genetic selection for LRFI reduces the total tract digestive capacity of pigs challenged with bacterial endotoxin [30], a significant contribution of the digestive tract to Trp postprandial utilization seems unlikely since Trp is not extensively used by the gut [31]. To summarize, lower average concentrations of Trp and Arg in LRFI pigs may result from an increased utilization of these two AA for immune purposes and may contribute to support the greater ability of LRFI pigs to cope with poor hygiene of housing conditions and to maintain their growth rate in challenging conditions [7]. When housed in poor hygiene conditions, HRFI pigs had greater average concentrations and maximum plasma concentration (Cmax) of Lys than those housed in good conditions whereas no difference between hygiene was reported in LRFI pigs. Greater plasma Lys concentrations are probably a consequence of lower body protein synthesis and reduced efficiency of nitrogen utilization for protein retention in pigs when the immune system is overstimulated [32]. Accordingly, Chatelet et al. [7] reported that ADG of the HRFI pigs was more affected by poor hygiene of housing conditions with a difference in ADG between poor- and good-HRFI pigs being twice the differences observed between poor- and good-LRFI pigs.

In conclusion, our results show that insulin, energy metabolites, and AA postprandial profiles differ between LRFI and HRFI pigs in response to poor hygiene of housing conditions. The findings that only LRFI had lower concentrations of Trp and Arg in poor hygiene of housing conditions may be associated with a greater AA utilization for supporting their immune responses. Conversely, only HRFI pigs had greater concentrations of Lys in poor hygiene of housing conditions, which could be in accordance with our previous study showing a greater impact of the challenge on average daily gain of HRFI pigs. Although our results clearly showed that selection for RFI modified the metabolic response to the hygiene challenge, it is not possible to determine whether selection has first modified the immune response or some metabolic pathways.

## Acknowledgments

The authors thank the staff of INRAE PEGASE and UE3P for their collaboration.

## Author Contributions

**Conceptualization:** Nathalie Le Floc'h.

**Formal analysis:** Nathalie Le Floc'h.

**Funding acquisition:** Nathalie Le Floc'h.

**Methodology:** Isabelle Louveau, Nathalie Le Floc'h.

**Project administration:** Nathalie Le Floc'h.

**Supervision:** Nathalie Le Floc'h.

**Validation:** Nathalie Le Floc'h.

**Writing – original draft:** Alícia Zem Fraga.

**Writing – review & editing:** Isabelle Louveau, Paulo Henrique Reis Furtado Campos, Luciano Hauschild, Nathalie Le Floc'h.

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
