## [Decision Letter · Decision Letter 0]

10 Feb 2021

PONE-D-21-00948

Selection for feed efficiency elicits different postprandial plasma metabolite profiles in response to poor hygiene of housing conditions in growing pigs

PLOS ONE

Dear Dr. Le Floch,

Thank you for submitting your manuscript to PLOS ONE, and sorry for the delay in evaluating your manuscript, that was due to the difficulty in finding expert with the relevant expertise. After careful consideration, we feel that it has merit but does not fully meet PLOS ONE’s publication criteria as it currently stands. Therefore, we invite you to submit a revised version of the manuscript that addresses all the points raised during the review process.

Particularly, you will see that the expert reviewer asked for additional explanations regarding several aspects of your study, as well as corrections regarding several inconsistencies and mistakes in the text of your manuscript.

We look forward to receiving your revised manuscript.

Kind regards,

Francois Blachier, PhD

Academic Editor

PLOS ONE

Journal Requirements:

2. In your Methods, please provide full details of animal care and housing.

3. Thank you for including your ethics statement:  "The experiment was conducted at INRAE UE3P (Saint-Gilles, France) in accordance with the ethical standards of the European Community (Directive 2010/63/EU), and was approved by the Regional ethical committee (CREEA number 07). The experiment approval number is APAFIS#494-2015082717314985.".   

Please amend your current ethics statement to include the full name of the ethics committee that approved your specific study.

For additional information about PLOS ONE submissions requirements for ethics oversight of animal work, please refer to http://journals.plos.org/plosone/s/submission-guidelines#loc-animal-research  

Reviewers' comments:

Reviewer's Responses to Questions

**Comments to the Author**

1. Is the manuscript technically sound, and do the data support the conclusions?

Reviewer #1: Partly

2. Has the statistical analysis been performed appropriately and rigorously? 

Reviewer #1: Yes

3. Have the authors made all data underlying the findings in their manuscript fully available?

Reviewer #1: Yes

4. Is the manuscript presented in an intelligible fashion and written in standard English?

Reviewer #1: Yes

5. Review Comments to the Author

Reviewer #1: Overall an interesting study with interesting results. There are however some major and minor changes to make before it can be published. Some explanations are missing and some inconsistencies need to be solved first.

Major remarks:

1. Be consistent in thermology, in the manuscript more different terms are used to indicate the same treatment. This is confusing and should be more consistent. For example; In the title you write ‘poor hygiene of housing conditions’, L19 ‘poor environmental conditions’, L31/32 ‘poor hygiene conditions’, 111 ‘health status’, L117 ’Hygiene challenge’ etc.

2. In the conclusion is stated (L415/416): …’which is in line with the greater impact of the challenge on protein deposition’. I don’t think this can be stated as there is no significant effect on ADG of the pigs in this experiment. So, this should be deleted.

3. In the manuscript several times the word ‘hyper-activation’ is used. I think ‘hyper’ is suggesting the wrong thing, such as over-activation. Activation without hyper will do.

4. L59 this sentence suggests that this study is already done and known. Make clear why /what is new in your study.

Minor remarks:

L25 (W3), and

L43 delete ‘the’ at ‘with the poor hygiene’…

L47 you write; ‘growth and maintenance’ but shouldn’t it we maintenance only? The next sentence explains it by heat production, physical activity and metabolism, so I understand only directly on maintenance and indirectly on growth.

L53 ‘like’ should be ‘such as’

L56 results

L56/57 delete ‘to support the immune responses’

L57 reduces

L63 add ‘,and nitrogen- and energy-related’

L64 metabolites, as it

L79 body weight (BW)

L84 95 kg BW

L91-93 How were cleaning, and disinfection done, and precautions taken?

L98 very brief information of the diet, make it more detailed. L98 complete soya beans or soy bean meal? What time of bran? Ricebran?

L99 meet or exceed is quite vague. Perhaps it is better to at a table with the exact diet.

L101 L-methionine or DL-methionine?

L113-L116 very long sentence

L120 15% mealsize? On average for all pigs? Maintenance energy for poor might be different than for good conditions.

L121 ADFI

L114 There is written ‘assay sensitivity’ but it is not clear to what assay it is referring to.

L189-190 No effect on ADG? Explain in discussion what can be the reason. A difference in ADG is expected right? I would expect a difference in ADG especially in the starter phase (25 kg-60kg). 6 weeks is also long enough to see a clear effect normally. What about an effect on ADG for LRFI of HRFI pigs? This is not presented but should at least be written in the results.

L194, L196 Be consistent with p-values P<0.05 or p = 0.04, both significant why showing them in a different way?

L197 write the letters of the figure in capitals, as is done in the figures.

Table 1. and other tables. Is it an option to show non-significant values as n.s.?

L295 nitrogen-related metabolites comes out of the blue for me, please explain also in introduction.

L297 Briefly, after being fasted overnight, on the day of serial blood sampling, all pigs….

L303 how do you know this is inflammation? Was fever measured?

L303 ‘health’ this term is difficult to use here. With an activated immune system you can be perfectly healthy….

L304 ‘decreases pig performance’, this was not the case in the current study. No effect on ADG was found for poor or good conditions.

L307 ‘inflammation’… I would suggest to use immune activation and not inflammation. No body temperature was measured.

L308 overstimulated should be stimulated

L316 ‘showing the fast return to glucose homeostasis’ or a too low challenging effect…?

L319 lower AA retention as muscle protein, if so an effect on ADG should be present and this was not the case…

L410 metabolites

L414 ‘and growth’ should be deleted (no effect on ADG was found…)

L416 Impact of challenge on protein deposition? This cannot be stated as there is no data supporting a difference. Or you have to show data with difference for protein deposition of difference in ADG.

6. PLOS authors have the option to publish the peer review history of their article (what does this mean?). If published, this will include your full peer review and any attached files.

Reviewer #1: No

---

## [Author Response · Author response to Decision Letter 0]

26 Feb 2021

Dear Editor,

We greatly thank the reviewer for the critical and helpful comments on our manuscript that significantly contributed to this new version. As you will see, we carefully addressed each of the reviewer’s comments and we hope that all changes will meet your expectations.

Our answers are listed in the covering letter and a version of the revised manuscript with all the modifications (i.e., corrections and additional information inserted) highlighted in red is provided (manuscript with track changes).

Yours sincerely,

Nathalie Le Floc’h

---

## [Decision Letter · Decision Letter 1]

16 Mar 2021

Selection for feed efficiency elicits different postprandial plasma metabolite profiles in response to poor hygiene of housing conditions in growing pigs

PONE-D-21-00948R1

Dear Dr. Le Floch,

We’re pleased to inform you that your manuscript has been judged scientifically suitable for publication and will be formally accepted for publication once it meets all outstanding technical requirements.

Kind regards,

Francois Blachier, PhD

Academic Editor

PLOS ONE

Additional Editor Comments (optional):

Reviewers' comments:

Reviewer's Responses to Questions

**Comments to the Author**

1. If the authors have adequately addressed your comments raised in a previous round of review and you feel that this manuscript is now acceptable for publication, you may indicate that here to bypass the “Comments to the Author” section, enter your conflict of interest statement in the “Confidential to Editor” section, and submit your "Accept" recommendation.

Reviewer #1: All comments have been addressed

2. Is the manuscript technically sound, and do the data support the conclusions?

Reviewer #1: Yes

3. Has the statistical analysis been performed appropriately and rigorously? 

Reviewer #1: Yes

4. Have the authors made all data underlying the findings in their manuscript fully available?

Reviewer #1: Yes

5. Is the manuscript presented in an intelligible fashion and written in standard English?

Reviewer #1: Yes

6. Review Comments to the Author

Reviewer #1: (No Response)

7. PLOS authors have the option to publish the peer review history of their article (what does this mean?). If published, this will include your full peer review and any attached files.

Reviewer #1: **Yes: **Yvonne van der Meer, PhD

---

## [Editor Report · Acceptance letter]

18 Mar 2021

PONE-D-21-00948R1 

Selection for feed efficiency elicits different postprandial plasma metabolite profiles in response to poor hygiene of housing conditions in growing pigs 

Dear Dr. Le Floc’h:

I'm pleased to inform you that your manuscript has been deemed suitable for publication in PLOS ONE. Congratulations! Your manuscript is now with our production department. 

Kind regards, 

on behalf of

Dr. Francois Blachier 

Academic Editor

PLOS ONE